# Response of Soil Microbial Community Structure and Diversity to Mixed Proportions and Mixed Tree Species in Bamboo–Broad-Leaved Mixed Forests

**Meiman Zhang [1,2,*], Fengying Guan [2], Shaohui Fan [2] and Xuan Zhang [2]**

[1]    College of Life and Environmental Science, Huangshan University, Huangshan 245041, China
[2]    State Key Lab for Bamboo and Rattan Science, International Center for Bamboo and Rattan, Beijing 100102, China
*    Correspondence: 600073@hsu.edu.cn; Tel.: +86-188-5597-3619

**Abstract:** Bamboo and broad-leaved mixed forests have been widely recognized for their advantages in maintaining ecological balance, improving soil fertility, and enhancing biodiversity. To understand the effects of mixed broad-leaved tree species and mixing ratios on soil microbial communities in bamboo and broad-leaved mixed forests, we quantified the structure and diversity responses of soil microbial communities to tree species and mixing ratios using high-throughput sequencing of the 16 S rRNA gene. Three bamboo and broad-leaved tree mixed forests were studied, including bamboo–*Castanopsis chinensis* Hance mixed forest (CCB), bamboo–*Alniphyllum fortune* (Hemsl.) Makino mixed forest (AFB), and bamboo–*Choerospondias axillaris* (Roxb.) B. L. Burtt & A. W. Hill mixed forest (CAB). We assessed the impact of tree species and mixing ratios on soil microbial communities by measuring soil properties and the diversity and composition of soil microbes. The results indicate that soil properties and the diversity and composition of microbial communities are highly dependent on broad-leaved tree species in mixed forests. The mixing ratios had a more pronounced effect on microbial diversity than on richness. In CAB, diversity peaked at mixing ratios of 10%–20% and 20%–40%. The presence of broad-leaved trees significantly altered the relationships among soil bacteria, with CAB showing the highest stability, likely due to the increased diversity and quantity of litter from *Choerospondias axillaris*. Our results show that the choice of broad-leaved tree species and their mixing ratios significantly influence soil microbial diversity and composition in bamboo–broad-leaf mixed forests. An optimal mixing ratio in CAB can maximize bacterial diversity and stability, providing insights for forest management and promoting ecosystem health and sustainability.

**Keywords:** bamboo–broad-leaved mixed forest; mixed proportion; *Choerospondias axillaris* (Roxb.) B. L. Burtt & A. W. Hill; *Alniphyllum fortune* (Hemsl.) Makino; *Castanopsis chinensis* Hance; soil microbial community and diversity

## 1. Introduction

Trees affect the diversity, community structure, and functionality of soil microorganisms through several pathways, most importantly through their root exudates, the quality and quantity of their litterfall, and the unique ecological characteristics of the species [1–3]. Soil microorganisms are often limited by carbon, and a decrease in rhizosphere carbon input will result in a decrease in the number of symbionts utilizing the unstable carbon matrix and an increase in the number of oligotrophics adapted to carbon-poor environments [4,5]. Different tree species have significantly different carbon resources in their litterfall [6,7], which leads to variations in the nutrients available for microbes. These differences alter the proliferation of soil microorganisms and directly affect the composition and activity of the soil microbial community. The competitive advantage of broad-leaved trees compared with conifers in mixed forest leads to a decrease in the rhizosphere biomass allocation of broad-leaved trees in mixed forest and a decrease in the subsequent input of rhizosphere

carbon to rhizosphere soil compared with pure stands [8,9], and the roots of trees and mycorrhizal fungi also affect soil carbon storage [4,5]. Exploring the interactive mechanisms between forest trees and soil microbial communities may allow clarification of how trees regulate the compositional dynamics of microbial community structures.

Bamboo is cultivated extensively worldwide because of its versatility [10,11]. Over the past few decades, the coverage of Moso bamboo (*Phyllostachysheterocycla* (carr.) Mitford cv. *Pubescens*) forests in China's subtropical regions has increased rapidly at an average annual rate of 3 percent [12,13]. Although it provides wood resources and vegetables to humans and possesses a short growth cycle, green safety, and strong material flexibility, Moso bamboo can have negative impacts on soil microorganisms, nutrient cycling, and ecological functions. Shen et al. (2015) [14] and Guo et al. (2016) [15] found that the abundance of nitrogen-fixing bacteria and anaerobic bacteria in bamboo forests were significantly lower than that in broad-leaf. To improve microbial communities, an empirical model of mixed bamboo and broad-leaved forest was proposed because it provides a diverse range of litterfall, increasing the nutritional sources for soil microorganisms [16]. Further, this mixed-species planting model enhances the cycling of carbon [17] and nutrients in the soil, contributing to the construction of more stable and efficient ecosystem service functions [18]. As a kind of ecological forestry management model, bamboo and broad-leaved mixed forest has been widely recognized for its advantages in maintaining ecological balance, improving soil fertility, and enhancing biodiversity. However, it is not well understood how the effects of bamboo and broad-leaved mixed forest on soil microorganisms vary with changes in broad-leaved tree species and mixing ratios.

The effects of tree species type and mixing ratio on soil microbial communities are important topics in forest ecology. The composition of tree species has a significant effect on soil microbial communities and composition, which has been confirmed in a number of studies [19,20]. The mixed-species forestry model enhances the diversity of litterfall types and quantities by incorporating different tree species, thereby providing soil microorganisms with a varied nutritional source and habitat [21,22]. This diversity not only enriches the ecological niches available to soil microbes but may also influence the microbial community structure and activity by altering the soil's physical and chemical properties, such as increasing soil porosity and organic matter content [23,24]. The adjustment of mixing proportions further refines this impact; an appropriate proportion of mixing can maximize the complementary effects between different tree species, promoting an increase in microbial diversity and the enhancement of ecosystem functions [25]. Li et al. (2003) [26] successfully established a mixed planting of bamboo with other trees at the optimal density level, ensuring sufficient space and light for the cultivation of a sustainable proportion of broad-leaf trees and understory vegetation. There is a significant difference in the diameter at breast height (DBH) between the bamboo and the broad-leaved trees in bamboo and broad-leaved tree mixed forests. It is noteworthy that once the bamboo reaches maturity, its DBH does not change with the passage of time. Additionally, the broad-leaved trees that coexist with the bamboo in these mixed forests often have a taller stature and larger crowns, which have a significant impact on the microclimate and immediate environment of the forest floor. Therefore, employing the appropriate method to determine the mixing ratio is crucial for revealing the complex mechanisms by which mixed forests influence soil microbial communities. This has significant theoretical and practical implications for optimizing forest ecosystem management, improving soil quality, and promoting biodiversity conservation.

The aim of this study was to investigate the effect of mixed broad-leaved tree species and mixing ratios on soil properties, soil microbial communities, and soil bacterial diversity in bamboo and broad-leaved mixed forests. We hypothesized that (1) AFB will have a greater adverse effect on soil nutrients and microbial structure than CAB, and an intermediate effect compared to CCB. Further, we hypothesized that (2) microbial structure and diversity will be strongly impacted by mixed forest types in intermediate-mixing-ratio forests, but less so in lower-mixing-ratio forests; we also hypothesized that (3) the mixed proportion and mixed species of bamboo and broad-leaved mixed forests affect microbial

community structure and diversity by influencing soil nutrient status. We assessed the impact of tree species and mixing ratios on soil microbial communities by measuring soil properties and the diversity and composition of soil microbial communities. Three bamboo and broad-leaved tree mixed forests were designed to put these hypotheses to the test, including bamboo–*Castanopsis chinensis* Hance mixed forest (CCB), bamboo–*Alniphyllum fortune* (Hemsl.) Makino mixed forest (AFB), and bamboo–*Choerospondias axillaris* (Roxb.) B. L. Burtt & A. W. Hill mixed forest (CAB).

## 2. Methods

### 2.1. Site Description

The study area was located in the Tianbaoyan Nature Reserve (TNR; 117°28′03″ E–117°35′28″ E; 117°28′03″ E–117°35′28″ E) at the junction of the Xiyang, Shangping, and Qingshui Townships (towns) in the eastern part of Yong'an City, Fujian Province in China. This protected area is within a low-relief landform known as the Daiyunshan vestibule. The main type of soil is red earth, lying at 580–1604.8 m a.s.l. This area has a medium subtropical southeast monsoon climate, with an annual average temperature of 15 °C (min. and max. are −11 °C and 40 °C, respectively), an annual average relative humidity exceeding 80%, and an annual average frost-free period lasting ca. 290 days [27]. The coverage rate of bamboo forests in protected areas is currently 96.8% and is mostly distributed below 800 m a.s.l. Moso bamboo is the predominant bamboo species. The main tree species co-occurring with Moso bamboo are *Choerospondias axillaris* (Roxb.) B. L. Burtt & A. W. Hill, *Castanopsis fargesii* Franch., *Alniphyllum fortune* (Hemsl.) Makino, *Cinnamomum camphora* (L.) Presl., *Liquidambar formosana* Hance, *Sassafras tzumu* (Hemsl.) Hemsl., and *PhoebezhennanS.* Lee; other species include *Schima superba* Gardner & Champ., *Quercus chenii* Nakai, *Myrica rubra* (Lour.) S. et Zucc., and *Castanopsis chinensis* Hance.

### 2.2. Filed Sites and Soil Sampling

The study included three forest types arranged as a trio with three mixing ratios covering a range of environmental conditions (3 forest types × 3 mixing ratios). We focused on the unique coexistence pattern formed by bamboo forests and three large deciduous broad-leaf tree species, a pattern that is clearly discernible within the study area. Specifically, this distinctive ecosystem is composed of bamboo intermingled with three particular tree species: *Castanopsis chinensis* Hance (CC), *Alniphyllum fortune* Hemsl. Makino (AF), and *Choerospondias axillaris* (CA). Based on this composition, we categorized such forestlands into three types: Bamboo–*Castanopsis chinensis* Hance mixed forest (CCB), Bamboo–*Alniphyllum fortune* Hemsl. Makino mixed forest (AFB), and Bamboo–*Choerospondias axillaris* (Roxb.) B. L. Burtt & A. W. Hill mixed forest (CAB). Within each type, the crown diameters of the deciduous tree species are similar, ranging approximately between 9 and 13 m, and only bamboo grows within a 9 m radius around each individual tree.

To more thoroughly investigate the characteristics of these mixed forests, we further subdivided the forest according to the bamboo–tree mixing ratio into three levels: 0%–10%, 10%–20%, and 20%–40% (calculated as the proportion of the vertical projection of the deciduous tree canopy coverage to the total plot area, with circular plots centered on the deciduous tree, with a radius of 10 m). At each level, three independent plots were established for each tree species as replicate observation points, and a total of 27 plots were established; the minimum distance between plots was 50 m to avoid pseudo replication. The absolute height of the studied sites ranged between 710 and 816 m (Supplementary Tables S1–S3); the slope direction of studied sites was sunny.

We focused on the area under the canopy shade cover (0–6 m away from the trunk) for each plot, conducting meticulous soil sampling. During sampling, we used a metal cylinder to extract soil cores with a diameter of 5 cm, taking eight cores each from the uphill, downhill, and horizontal directions relative to the trunk, which were then combined to form a single soil sample. Each sample weighed 100 g, and we ensured that the sampling depth corresponded to the 0–10 cm surface soil layer after the removal of leaf litter. Once

collected, after quartering, all soil samples were immediately divided into three portions, with two portions quickly refrigerated and taken to the laboratory for storage at $-80\,°C$ and $4\,°C$ for subsequent DNA, soil ammonium nitrogen ($NH_4^+$), and nitrate nitrogen ($NO_3^-$) analysis. The other portion was air-dried at the laboratory, its impurities were removed (e.g., large pieces of plant material, gravel, earthworms), and then the sample was ground. Non-refrigerated samples were crushed, sifted, and sealed in bags for later determination of soil nutrient content and enzyme activity [27].

### 2.3. Soil Chemical Analysis

Soil pH was determined using an electrode pH meter (Sartorius PB-10, made in Göttingen, Germany), with a soil-to-water ratio of 1:2.5. Soil organic carbon (SOC) was analyzed using a potassium dichromate–sulfuric acid solution and an external oil bath heated for oxidation [28]; DOC was determined with a total organic carbon analyzer (multi N/C 2100 TOC, Analytik Jena, Jena, Germany); and total nitrogen (TN), ammonium nitrogen ($NH_4^+$), nitrate nitrogen ($NO_3^-$), and total phosphorus (TP) were determined using an elemental analyzer (Smartchem 200, AMS Alliance, Rome, Italy). Potassium in the soil was primarily determined by flame photometry; soil alkali-hydrolyzable nitrogen (AN) was determined using a diffusion dish method [29], while a spectrophotometer was employed for the measurement of available phosphorus (AP) in the soil [30].

### 2.4. Soil Microbial Community

(1) DNA Extraction: Total bacterial DNA was extracted from each soil sample using a commercial soil microbial DNA extraction kit (MO BIO Laboratories, Carlsbad, CA, USA) according to the manufacturer's instructions. The quality and quantity of the extracted DNA were evaluated based on the absorbance ratios at 260 nm/280 nm and 260 nm/230 nm, respectively. The DNA was then stored at $-80\,°C$ until further processing [31].

(2) PCR Amplification: Bacterial gene amplification was performed using widely employed primers targeting the V3–V4 region (Xi et al., 2019 [32]; Wen et al., 2019 [33]): the forward primer 5′-ACTCCTACGGGAGGCAGCA-3′ and reverse primer 5′-GGACTACHVGG GTWTCTAAT-3′. The total volume of the PCR mixture was 50 μL, containing 25 μL of buffer (KOD FX Neo Buffer (2×), 1 μL of KOD FX Neo DNA polymerase (TOYOBO, Osaka, Japan), 10 μL of 2 mM dNTPs, 1.5 μL of each primer (Vn R (10 μM)), approximately 60 ng of genomic DNA, and ddH$_2$O to bring the final volume to the desired level. The PCRs were carried out on a 96-well PCR machine (PCR 9902, AB, Waltham, MA, USA) under the following thermocycling conditions [32]:

$$15 \text{ cycles} \begin{cases} 95\,°C & 5 \text{ min} \\ 95\,°C & 1 \text{ min} \\ 50\,°C & 1 \text{ min} \\ 72\,°C & 1 \text{ min} \\ 72\,°C & 7 \text{ min} \end{cases}$$

Purification of the first-round PCR products was carried out using VAHTSTM DNA Clean Beads (Vazyme.N411-01, Nanjing, China) [32]. The second-round PCR was performed as a 40-cycle polymerase chain reaction (PCR), comprising 20 μL of PCR mix (2× Phμsion HF MM), 8 μL of ddH$_2$O, 2 μL of each primer (10 μM) for PCR, and 10 μL of the purified product from the first-round PCR. The thermal cycling conditions were as follows: initial denaturation at $98\,°C$ for 30 s, followed by 10 cycles of denaturation at $98\,°C$ for 10 s, annealing at $65\,°C$ for 30 s, extension at $72\,°C$ for 30 s, and a final extension step at $72\,°C$ for 5 min [32].

(3) Sequence preprocessing: Based on the overlapping characteristics present among the paired-end reads, the dual-end reads obtained from the Illumina HiSeq platform were subjected to filtering and assembly, resulting in optimized sequences (Clean tags). Utilizing the UCLUST algorithm within QIIME software version v.1.8.0 [34], clustering was con-

ducted at a 97% similarity level, generating corresponding OTUs (Operational Taxonomic Units). Each OTU was subsequently classified according to the Silva taxonomy database.

### 2.5. Statistical Analyses

All statistical analyses were performed using SPSS (version 20.0) and R with the "Vegan" package. IBM SPSS Statistics was employed for two-factor ANOVA and multiple comparisons (LSD, $p < 0.05$) on soil nutrients and other variables. Intra-forest soil bacterial community evenness and diversity were calculated based on OTUs, while changes in the Beta diversity of the bacterial communities were analyzed using the "Vegan" package in R. Principal Coordinate Analysis (PCoA) was conducted based on Bray–Curtis dissimilarity to investigate the structure of the forest soil microbial community; clustering analysis, also utilizing the Bray–Curtis algorithm, was performed to elucidate relationships between bacterial communities in different forest stands. Venn diagrams were generated to illustrate shared and unique OTUs across different forest types. Redundancy Analysis (RDA) was used to assess the correlations between soil physicochemical properties and the microbial community. Molecular Ecological Network Analysis (MENA) was conducted on the platform at http://ieg4.rccc.ou.edu/mena/ (accessed on 15 January 2023). PCoA and RDAs were carried out in R, with graphical outputs generated using Excel 2016, Origin 9.0, and R software version 4.0.2.

## 3. Results

### 3.1. Changes in Soil Properties

All soil parameters were significantly different among different stands (Table 1). The mixed broad-leaved tree species and their individual sizes (mixed proportion) had a significant influence on the pH. The lowest pH was observed in the CAB stand (4.41), and the soil pH value decreased with an increase in the crown width of *Choerospondias axillaris.* The highest SOC (46.50 g/kg), TN (2.69 g/kg), TP (0.497 g/kg), AP (5.22 mg/kg) and AK (59.64 mg/kg) contents were observed in the CAB stands with a 10%–20% mixing ratio. The DOC was higher in the CAB stands than the AFB and CCB stands, and it was most pronounced in the forest with a mixing ratio of 20%–40% (201.67 mg/kg). Compared with *Alniphyllum fortune* (Hemsl.) Makino, *Choerospondias axillaris* and *Castanopsis chinensis* Hance reduced the content of $NO_3^-$, $NH_4^+$, and TK. The $NO_3^-$ in CCB stands was gradually higher than that in CAB stands with an increase in the crown width of broad-leaved trees. The highest $NH_4^+$ and TK contents were observed in the AFB, with mixing ratios of 10%–20% (11.97 mg/kg) and 0%–10% (35.91 g/kg).

### 3.2. Effects of Mixed Tree Species and Mixing Ratios on the Soil Bacterial Community Composition

The soil bacterial community was dominated by *Acidobacteria* (29.56%–38.71%), *Proteobacteria* (25.02%–30.96%), and *Actinobacteria* (10.26%–15.98%); *Chloroflexi, Planctomycetes, Verrucomicrobia, Gemmatimonadetes, Firmicutes,* and *Bacteroidetes* were the main dominant bacteria, and the cumulative average relative abundance reached more than 95% (Supplementary Table S4). *Acidobacteria, Proteobacteria,* and *Actinobacteria* were the three dominant phyla in all forest sites, and significant differences were observed among forest types, with the highest relative abundance observed in the CAB and AFB stands and the lowest relative abundance observed in the CCB stands (Figure 1). When the effect of the mixing ratio on dominant bacteria was considered, significant differences were observed in the *Acidobacteria* and *Actinobacteria* among the mixing ratios in CAB, and the relative abundance of *Acidobacteria* and *Actinobacteria* was significantly increased with an increase in the mixing ratios (Figure 2).

**Table 1.** Effects of tree species and mixing ratio in bamboo and single broad-leaved tree mixed forests on soil nutrients.

| Nutrient Index | AFB | | | CAB | | | CCB | | |
|---|---|---|---|---|---|---|---|---|---|
| | 0%–10% | 10%–20% | 20%–40% | 0%–10% | 10%–20% | 20%–40% | 0%–10% | 10%–20% | 20%–40% |
| pH | 4.63 ± 0.12 Ab | 4.93 ± 0.06 Aa | 4.71 ± 0.03 Aab | 4.80 ± 0.04 Aa | 4.51 ± 0.05 Cb | 4.41 ± 0.02 Cb | 4.64 ± 0.02 Aa | 4.74 ± 0.05 Ba | 4.53 ± 0.02 Bb |
| SOC (g/kg) | 23.87 ± 0.22 Ca | 25.36 ± 0.19 Ca | 22.08 ± 0.08 Cc | 35.61 ± 0.65 Ab | 46.50 ± 0.28 Aa | 31.32 ± 0.08 Ac | 27.73 ± 0.19 Bb | 30.83 ± 0.16 Ba | 27.43 ± 0.15 Bb |
| DOC (mg/kg) | 134.67 ± 5.51 Ba | 128.33 ± 6.66 Aab | 119.33 ± 4.16 Bc | 168.33 ± 7.23 Ab | 126.33 ± 4.93 Bc | 201.67 ± 9.61 Aa | 134.33 ± 6.03 Ba | 111.33 ± 4.62 Bb | 116.67 ± 4.93 Bb |
| TN (g/kg) | 1.82 ± 0.07 Ca | 1.89 ± 0.004 Ca | 1.79 ± 0.09 Ba | 2.44 ± 0.05 Ab | 2.69 ± 0.05 Aa | 2.31 ± 0.08 Ab | 2.08 ± 0.03 Bab | 2.189 ± 0.08 Ba | 1.99 ± 0.02 Bb |
| $NO_3^-$ (mg/kg) | 84.53 ± 6.19 Aa | 13.37 ± 1.30 Bc | 56.30 ± 1.51 Ab | 6.75 ± 0.58 Bb | 15.13 ± 1.15 Ba | 16.93 ± 1.53 Ca | 12.50 ± 1.15 Bc | 28.77 ± 2.50 Ab | 39.70 ± 11.97 Ba |
| $NH_4^+$ (mg/kg) | 6.78 ± 0.62 Aab | 11.97 ± 0.97 Aa | 6.55 ± 0.49 Bc | 5.56 ± 0.26 Ba | 4.39 ± 0.37 Ba | 3.84 ± 1.46 Aa | 5.30 ± 0.49 Ba | 5.20 ± 0.16 Bb | 6.98 ± 0.33 Aa |
| TP (g/kg) | 0.300 ± 0.008 Ab | 0.332 ± 0.002 Ba | 0.337 ± 0.004 Ba | 0.253 ± 0.001 Bc | 0.497 ± 0.003 Aa | 0.397 ± 0.006 Ab | 0.253 ± 0.002 Bb | 0.224 ± 0.002 Cb | 0.254 ± 0.002 Ca |
| AP (mg/kg) | 8.91 ± 0.36 Aa | 2.43 ± 0.09 Bc | 4.61 ± 0.09 Ab | 1.80 ± 0.05 Cc | 5.22 ± 0.16 Aa | 2.44 ± 0.09 Bb | 2.37 ± 0.15 Ba | 1.77 ± 0.10 Cb | 1.93 ± 0.20 Cb |
| TK (g/kg) | 35.91 ± 0.42 Aa | 30.07 ± 0.10 Ab | 35.60 ± 0.94 Aa | 24.96 ± 0.63 Cb | 30.43 ± 0.96 Aa | 22.87 ± 0.90 Bc | 28.67 ± 0.44 Ba | 20.67 ± 0.76 Bb | 13.64 ± 0.27 Cc |
| AK (mg/kg) | 46.05 ± 1.51 Aa | 46.90 ± 0.91 Ba | 41.19 ± 2.30 Bb | 40.89 ± 6.18 Ac | 59.64 ± 0.26 Aa | 51.01 ± 1.66 Ab | 48.10 ± 1.35 Aa | 32.12 ± 0.83 Cb | 51.29 ± 5.89 Aa |
| C:N | 13.14 ± 0.41 Ba | 13.43 ± 0.13 Ba | 12.39 ± 0.66 Aa | 14.59 ± 0.11 Ab | 17.29 ± 0.36 Aa | 13.57 ± 0.49 Ab | 13.32 ± 0.12 Aa | 14.14 ± 0.57 Ba | 13.82 ± 0.17 Aa |
| C:P | 79.67 ± 2.62 Ca | 76.43 ± 1.09 Ba | 65.59 ± 0.65 Cb | 140.66 ± 2.39 Ba | 93.48 ± 0.27 Cb | 79.01 ± 1.29 Bc | 109.46 ± 0.13 Ab | 137.62 ± 1.97 Aa | 107.96 ± 0.55 Ab |

Note: Uppercase letters indicate different tree species under the same ratio; lowercase letters indicate the difference under different ratios of the same tree species.

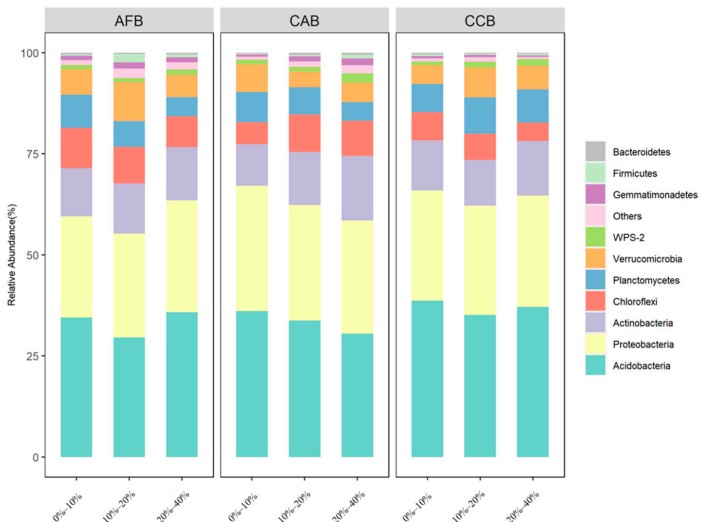

**Figure 1.** The community composition of dominant bacteria in bamboo and single broad-leaved tree mixed forest at the phylum level.

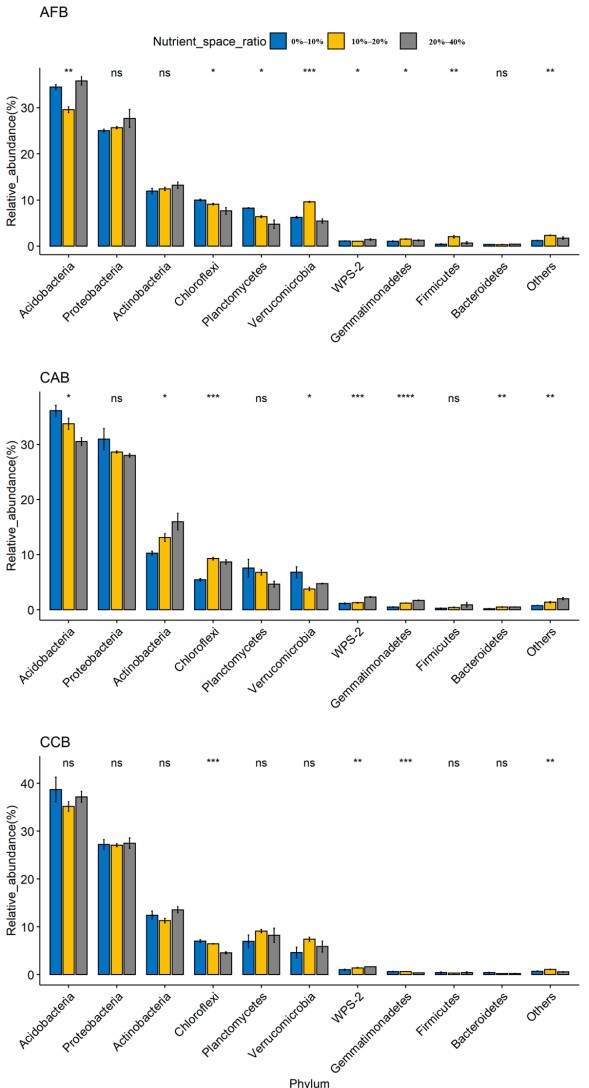

**Figure 2.** Relative abundances of the bacterial phyla. ns ($p > 0.05$): not significant; **** ($p < 0.0001$), *** ($p < 0.001$), ** ($p < 0.01$) and * ($p < 0.05$) indicate significant differences among forest types based on a one-way ANOVA followed by an LSD test.

In total, 983 species of OTUs were prevalent in this study. The interaction between mixed species and mixed proportions had no significant effect on the superposition of the soil bacterial communities (Figure 3). When the CAB stands possessed a mixing ratio of 20%–40%, the number of unique OTUs was 3. In AFB stands, the number of unique OTUs within the forest varied with different mixed proportions as follows: there were 5 unique OTUs present in the forest with a mixing ratio of 0%–10%; when the mixing proportion was between 10% and 20%, the number of unique OTUs increased to 12; and the number of unique OTUs decreased to 4 in the 20%–40% mixed proportion.

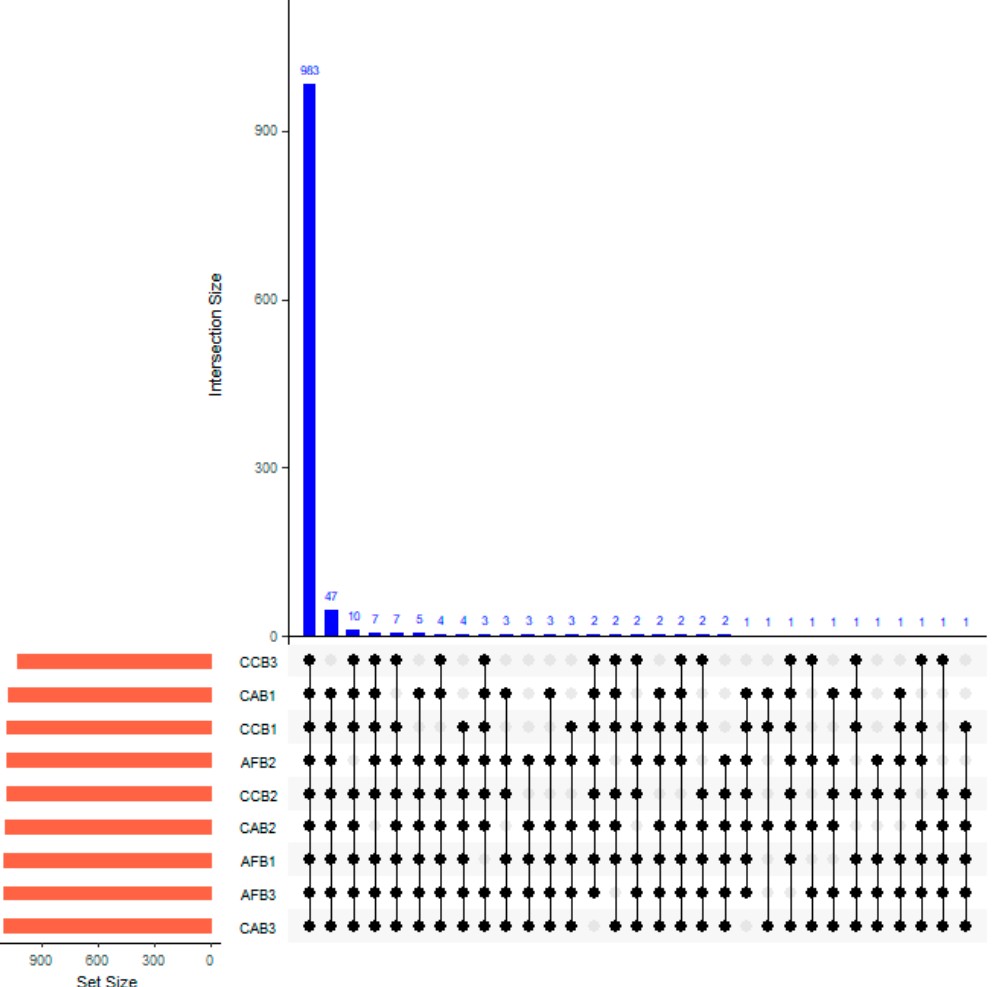

**Figure 3.** Venn diagram of unique soil and common bacteria OTUs (0.03 evolution distance) in bamboo and single broad-leaved tree mixed forest.

### 3.3. Effects of Mixed Tree Species and Mixing Ratio on the Soil Bacterial Diversity and Richness

The mixed proportion and mixed tree species had significant effects on the soil bacterial richness (ACE and Chao1 indexes) and diversity (Shannon and Simpson indexes) (Table 2). In bamboo–broad-leaved mixed forest, the highest soil bacteria diversity levels were observed in CAB stands with mixing ratios of 10%–20% and 20%–40%, and the lowest was soil bacterial richness were observed in CCB stands with a mixing ratio of 20%–40%.

With different mixed species and mixed proportions, the Beta diversity of the soil bacterial communities in each sample was significantly concentrated (Figure 4). The mixing ratio significantly changed the soil bacterial Beta diversity of CAB stands. The Beta diversity of soil bacterial communities in AFB stands was clustered together.

**Table 2.** Soil bacteria uniformity and diversity index in bamboo and single broad-leaved tree mixed forest.

| Nutrient Space Ratio | Tree Species | Soil Bacterial Richness | | Soil Bacterial Diversity | |
|---|---|---|---|---|---|
| | | ACE | Chao1 | Shannon_Index | Simpson_Index |
| 0%–10% | AFB | 1092 ± 3 Aa | 1096 ± 4 Aa | 5.83 ± 0.011 Aa | 0.0078 ± 0.0002 Aa |
| | CAB | 1060 ± 11 Aa | 1067 ±17 Aa | 5.69 ± 0.005 Ba | 0.0084 ± 0.0002 Ba |
| | CCB | 1063 ± 5 Aa | 1070 ± 11 Aa | 5.65 ± 0.057 Ba | 0.0099 ± 0.0007 Ba |
| 10%–20% | AFB | 1067 ± 10 Aa | 1075 ± 11 Aa | 5.834 ± 0.020 Aa | 0.0071 ± 0.0001 Aa |
| | CAB | 1079 ± 3 Aa | 1084 ± 4 Aa | 5.825 ± 0.016 Ab | 0.0081 ± 0.0003 Ba |
| | CCB | 1072 ± 3 Aa | 1076 ± 4 Aa | 5.65 ± 0.058 Ba | 0.0074 ± 0.0001 ABb |
| 20%–40% | AFB | 1177 ± 9 Aa | 1082 ± 10 Aa | 5.822 ± 0.0786 Aa | 0.0078 ± 0.0011 ABa |
| | CAB | 1089 ± 3 Aa | 1095 ± 4 Aa | 5.950 ± 0.037 Ac | 0.0058 ± 0.0005 Bb |
| | CCB | 990 ± 5 Bb | 998 ± 7 Bb | 5.65 ± 0.059 Ba | 0.0099 ± 0.0003 Aa |

Note: Capital letters indicate the difference between different tree species under the same ratio, and lowercase letters indicate the difference between different tree species under the same tree species.

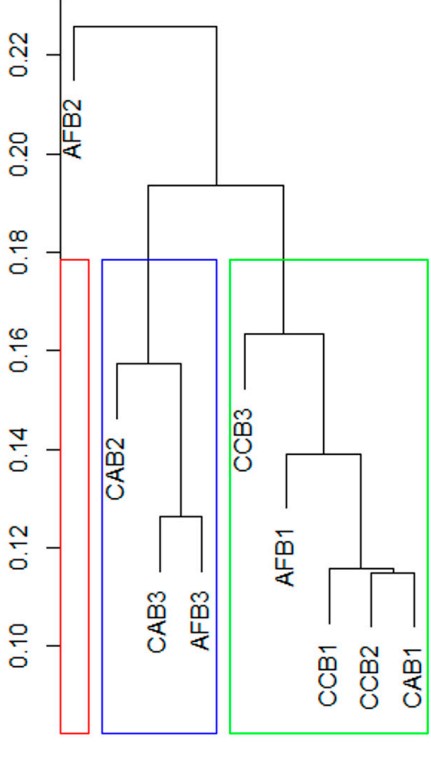

**Figure 4.** The clustering of soil bacteria in bamboo and single broad-leaved tree mixed forest.

*3.4. Effects of Mixed Tree Species and Mixing Ratio on the Structure of Soil Bacterial Communities*

A principal coordinate analysis was used to analyze the structure of soil bacterial communities; the first principal component explained 28.86% of the changes in soil bacterial community structure (species level) of the bamboo and broad-leaved mixed forests, and the second principal component explained 20.14% of the changes in the soil bacterial community structure (species level), accounting for 49% (Figure 5). The soil bacterial community structures in all bamboo and single broad-leaved tree mixed forests were mainly affected by mixing ratios, mixed tree species, and their interactions (Table 3).

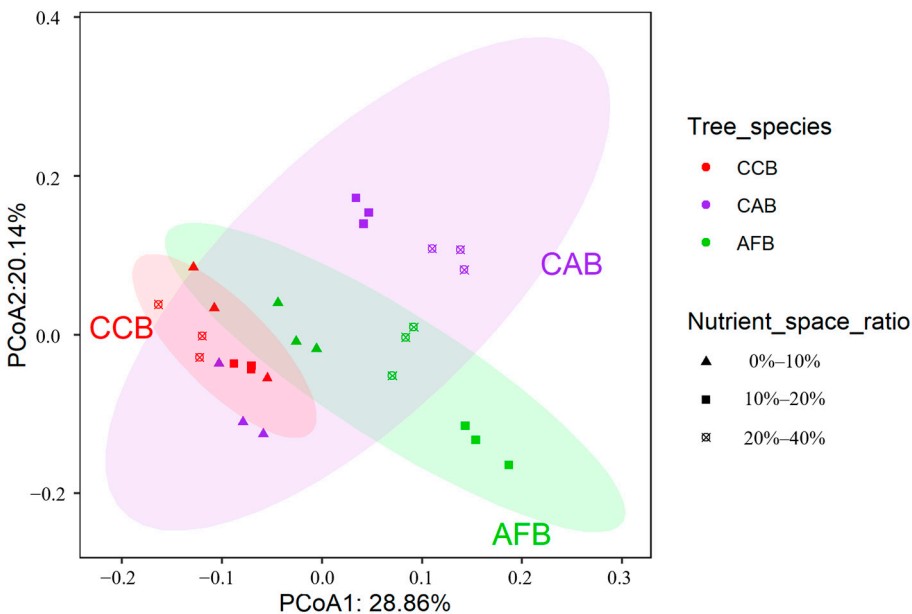

**Figure 5.** PCoA plot of soil bacterial communities from bamboo and single broad-leaved tree mixed forest based on the Bray–Curtis distance.

**Table 3.** ADONIS difference analysis of soil microbial community structure in bamboo and single broad-leaved tree mixed forest.

|  |  | Df | Sums of Squares | Mean Squares | F.Model | Variation ($R^2$) | Pr (>F) |
|---|---|---|---|---|---|---|---|
| Bacteria | Tree species | 2 | 0.2767 | 0.1383 | 4.2594 | 0.2620 | 0.001 |
|  | Residuals | 24 | 0.7794 | 0.0325 | 0.7380 |  |  |
|  | Total | 26 | 1.0561 | 1 |  |  |  |
| Bacteria | Mixing ratio | 2 | 0.1979 | 0.0990 | 2.7680 | 0.1874 | 0.003 |
|  | Residuals | 24 | 0.8581 | 0.0358 | 0.8126 |  |  |
|  | Total | 26 | 1.0561 | 1 |  |  |  |
| Bacteria | Tree species × Mixing ratio | 8 | 0.7929 | 0.0991 | 6.7800 | 0.7508 | 0.001 |
|  | Residuals | 18 | 0.2631 | 0.0146 | 0.2492 |  |  |
|  | Total | 26 | 1.0561 | 1 |  |  |  |

An analysis of soil microbial community structure networks was conducted in this study. The soil bacterial network was a scale-free network in all stands (Table 3). The soil bacterial molecular ecological network models in the CAB, AFB, and CCB stands had obvious small-world properties under different mixing ratios (empirical networks: avgCC 0.245–0.301, random networks: avgCC 0.017–0.05) (Table 4). The highest modular number was observed in the CAB stands. There were 13 module hubs and 4 connecting nodes in the AFB stands, 5 module hubs and 1 connecting node in the CAB stands, and 7 module hubs and 5 connecting nodes in the CCB stands (Figure 6).

**Table 4.** Topological characteristics of the empirical molecular ecological network of soil bacteria in bamboo and single broad-leaved tree mixed forest and random molecular ecological networks.

|  | Network Structure Attributes | AFB | CAB | CCB |
|---|---|---|---|---|
| Empirical networks | Similarity threshold | 0.940 | 0.960 | 0.950 |
|  | Total nodes | 628 | 479 | 508 |
|  | Total links | 2374 | 1038 | 1008 |
|  | $R^2$ of power law | 0.909 | 0.903 | 0.912 |

**Table 4.** *Cont.*

| Network Structure Attributes | | AFB | CAB | CCB |
|---|---|---|---|---|
| Empirical networks | Average path distance (GD) | 8.297 | 8.400 | 6.545 |
| | Average clustering coefficient (avgCC) | 0.301 | 0.273 | 0.245 |
| | Modularity and the number of modules | 0.590 (55) | 0.785 (56) | 0.741 (68) |
| | Module hubs | 13 | 5 | 7 |
| | Connectors | 4 | 1 | 5 |
| Random networks | Average path distance (GD) | 3.352 +/− 0.026 | 0.046 +/− 0.000 | 4.142 +/− 0.052 |
| | Average clustering coefficient (avgCC) | 0.050 +/− 0.005 | 0.019 +/− 0.003 | 0.017 +/− 0.003 |
| | Modularity (M) | 0.305 +/− 0.004 | 0.470 +/− 0.005 | 0.502 +/− 0.005 |

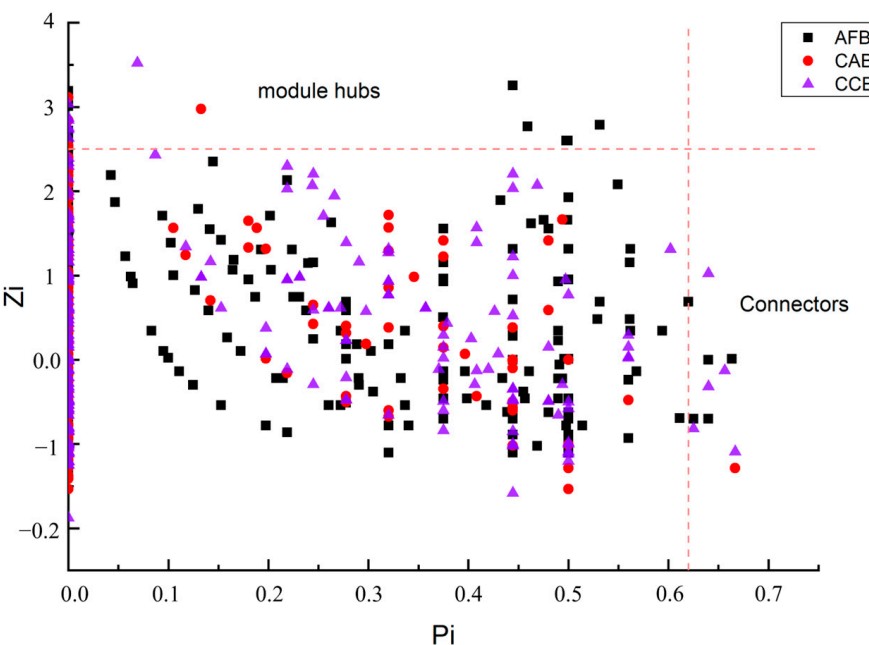

**Figure 6.** ZP plot showing distribution of OTUs based on their module-based topological roles.

*3.5. Effects of Soil Properties on the Composition of Soil Bacterial Communities*

The effects of soil properties on the composition of soil core bacterial communities at the species level were analyzed using RDA. The soil properties explained 68.74% of the total variation in the soil bacterial community composition at the species level following bamboo and single broad-leaved tree mixed forests with mixing ratios and tree species. The first two axes of the RDA explained 36.78% of the total variations in the core soil bacterial communities (Figure 7). The first axis of the RDA explained 21.08% of the total variability and was associated with DOC ($R^2$ = 0.485, $p$ = 0.003), TK ($R^2$ = 0.360, $p$ = 0.002), and TP ($R^2$ = 0.169, $p$ = 0.001); the second axis explained an additional 15.68% of the total variance and was associated with SOC ($R^2$ = 0.546, $p$ = 0.002), AK ($R^2$ = 0.420, $p$ = 0.004), and $NH_4^+$ ($R^2$ = 0.293, $p$ = 0.001).

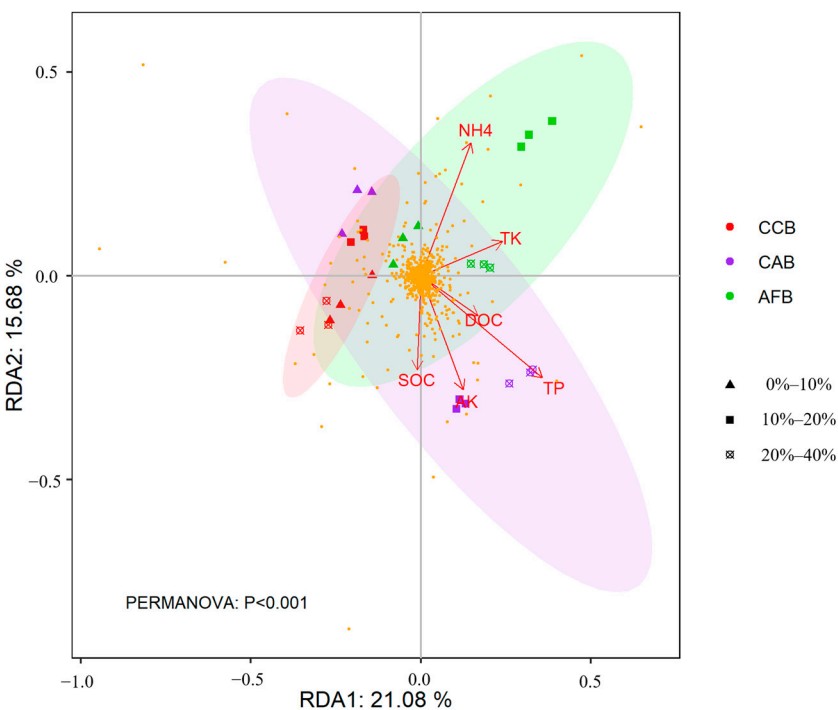

**Figure 7.** RDA showing the correlations between microbial community structure and soil factors.

## 4. Discussion

### 4.1. Effects of Mixed Tree Species and Mixing Ratios on Soil Properties

Bamboo and broad-leaved mixed forests with different tree species and mixing ratios significantly affect soil nutrients. In mixed forests, the canopy layer often has a great impact on forest productivity [34–36]. The differences in tree species and their canopy structures may significantly change the soil microenvironment in the forest, and greatly affect the physical, chemical, and biological characteristics of soil, especially the physical and chemical properties of soil around trees [37]. The soil pH values in AFB stands were significantly higher than that in CAB and CCB stands in this study, and the difference in soil pH value was obvious when different mixed proportions were analyzed, indicating that differences in broad-leaved tree species and broad-leaved canopy width had different effects on soil pH. In our study, the SOC and TN contents in the CAB stands were higher than those in the AFB and CCB stands, the lowest TP content was observed in the CCB stands, and the SOC and TN contents were higher in the AFB, CAB, and CCB stands with a mixing ratio of 10%–20%. This result may be partly explained by the difference in the amount and characteristics of soil litter input in the forest [38], which leads to a difference in the input of organic matter in the forest. In addition, the substrate of saprophytic organisms has an impact on soil fertility [39,40], and the quality of litters also affects its decomposition rate, resulting in a difference in the accumulation of organic matter. Kooch et al. (2016) [41] also reported that different tree species can strongly affect soil chemical properties. The soil DOC in different forest stands is relatively unstable, and vegetation types and litter have a certain degree of impact on their soluble nutrients. The content of DOC in soil is highest in CAB. The litter (fruit, leaves) of *Choerospondias axillaris* (Roxb.) B. L. Burtt & A. W. Hill has a certain impact on soil nutrients in the forest. The highest and lowest soil C and N concentrations were found in CAB and CCB, respectively. This may be due to *Castanopsis chinensis* Hance leaf litter being harder and its decomposition rate slower than that of *Alniphyllum fortune* (Hemsl.) Makino (AF) and *Choerospondias axillaris* (Roxb.) B. L. Burtt & A. W. Hill. The higher N content in *Choerospondias axillaris* (Roxb.) B. L. Burtt & A. W. Hill litter, followed by a more rapid decomposition [42], will improve the soil N in CAB compared to that of the other forests. Soil P concentrations in the CAB were higher than those in the CCB and AFB. Xu (2013) [43] showed that the P content

in the roots of *Choerospondias axillaris* (Roxb.) B. L. Burtt & A. W. Hill decreased during the root decomposition process, which indicated that the decomposition of fine roots of *Choerospondias axillaris* (Roxb.) B. L. Burtt & A. W. Hill released P and contributed greatly to soil nutrients.

### 4.2. Effects of Mixed Tree Species and Mixing Ratio on the Soil Bacterial Community

In bamboo and broad-leaf mixed forests, the diversity and richness of soil microbial community structure varied between different mixed forests under different mixing ratios. In this study, the ACE and Chao1 indexes revealed differences in the diversity and richness of soil microbial community structure across the three forest types, namely a richer bacterial community in CAB with 10%–20% and 20%–40% mixing ratios than in other forests; the lowest value was found in CCB with a 20–40% mixing ratio. This result may be attributed to the difference in the amount of light available with the change in stand structure, which affects the composition of understory plants [44,45] (Zhang et al., 2013; Cheng et al. 2016), and thus has a certain impact on the soil microbial diversity in the forest [46] (Rivest et al., 2019). *Choerospondias axillaris* (Roxb.) B. L. Burtt & A. W. Hill is a deciduous tree species whose light environment is more abundant than that of *Castanopsis chinensis* Hance (an evergreen tree species), which leads to a more diverse variety of sunny plants under the forest. Haghverdia (2019) [47] showed that as the species diversity, especially that of deciduous broad-leaved trees, increased, more nutrients were found to be deposited in the soil surface through litter [48]. In addition, the fruit of *Choerospondias axillaris* (Roxb.) B. L. Burtt & A. W. Hill is a good edible fruit, and there is a phenomenon of harvesting *Choerospondias axillaris* (Roxb.) B. L. Burtt & A. W. Hill fruit in the CAB site we studied. Liu et al. (2018) [19] reported that bacterial community diversity in forestland was lower than that in cropland. Jangid et al. (2008) [49] suggested that this could be attributed to human agricultural practices which changed the physical and chemical properties of soil, which altered soil bacterial diversity; thus, our results could have been further influenced by different *Choerospondias axillaris* (Roxb.) B. L. Burtt & A. W. Hill management practices that disturb surface soils, which could have contributed to increased soil bacterial diversity. The Beta diversity analysis showed that the soil bacterial community in AFB stands can gather together well under different mixing ratios, which indicated that the soil bacterial community in the AFB stand is more stable when compared with other forests. The PCoA analysis also confirmed the above conclusions, and these results indicate that the mixed ratio and mixed tree species play a crucial role in the formation of microbial communities.

*Acidobacteria*, *Proteobacteria*, and *actinomycetes* were the main dominant bacteria in our study; the relative abundance was highest in the CAB stands, and the difference was significant under different mixing ratios. Our study showed that the mixed species and the mixed proportion had a certain selectivity on the influence of soil bacteria composition in bamboo and broad-leaved mixed forests. A large number of studies have shown that *Acidobacteria*, *Proteobacteria*, and *Actinomyces* in soil can represent the soil nutritional status [50–52]. This study found that the relative abundance of *Acidobacteria*, *Proteobacteria*, and *actinomycetes* varied with different mixes of species and mixed proportions, which could mainly be attributed to the influence of soil nutrient status and soil microclimate in the forest. Different tree species lead to differences in litter content and characteristics within the forest. *Choerospondias* and *Castanopsis chinensis* Hance litter contains seeds, which increases litter content under the forest, leading to changes in soil microclimate and thus changes in soil microbial community structure.

Our study constructed molecular ecological networks of soil bacteria in different types of bamboo–broad-leaved mixed forests to better explain the response mechanisms of soil bacteria to mixed tree species and mixing ratios. Some topological characteristics of soil bacterial molecular ecological network models were studied from the perspective of the inter-relationship between soil bacteria, and the response of the models to mixed tree species and mixing ratios was analyzed. Mixed species and mixed proportions had significant effects on the structure of soil bacterial networks, and the stability of the soil

bacterial community structures was significantly changed by different broad-leaved tree species in the forest. The total numbers of nodes and the total numbers of connections in the molecular networks of soil bacterial communities in mixed broad-leaved–bamboo forests were significantly different between different species, which may be attributed to the differences in competition between bamboo and broad-leaved trees in the forest caused by different tree species and the differences in aboveground nutrient spatial distribution. Aboveground biomass is generally related to the nutrients and energy provided by underground roots. Subsurface microorganisms respond to environmental differences by competing and cooperating [53].

*4.3. Effects of Soil Properties on the Soil Bacterial Community*

The changes in the soil bacterial community composition in bamboo and broad-leaved mixed forests with different mixed tree species and mixing ratios were driven by the phyla *Acidobacteria*, *Proteobacteria*, *actinomycetes*, and *Bacteroidetes*, which were significantly affected by soil parameters, such as soil SOC, TP, TK, AP, DOC, $NH_4^+$, and pH. *Acidobacteria* are widely distributed in farmland and forest soils [54,55], and have been identified as oligotrophic bacteria, which are multi-purpose heterotrophs [56]. *Acidobacteria* have a slow metabolic rate under low-nutrient conditions, whereas *Proteobacteria* were identified as eutrophic bacteria and congregate in nutrient-rich soils, especially where carbon sources are abundant. *Proteobacteria* play an important role in the transformation and decomposition of soil nutrients, and have a positive effect on the growth of vegetation in a forest, especially the growth of bamboo [57]. *Actinomycetes* are greatly affected by soil water content and regional microclimate, which can promote the degradation of cellulose and lignin; therefore, it is conducive to the decomposition of litter under the forest and the soil, and promotes an increase in soil nutrition in the forest. *Bacteroidetes* facilitate the decomposition of soil organic matter, enhancing carbon mineralization and the release of carbon dioxide, while also participating in nitrogen fixation, mineralization, and nitrification processes, thereby increasing the availability of nitrogen in the soil. Additionally, *Bacteroidetes* contribute to phosphorus availability by secreting organic acids that enhance phosphorus solubility. *Bacteroidetes* activities not only bolster soil fertility but also critically support the overall productivity and stability of forest ecosystems.

Soil properties explained 68.74% of the total variation in the soil bacterial community composition at the species level following bamboo and single broad-leaved tree mixed forests with mixing ratios and tree species. The effect of DOC, $NH_4^+$, and SOC on soil bacterial community composition has been well documented in many other studies conducted at various scales [25,27,58]. For example, Huang et al. (2022) [25] studied the relationship between soil chemical properties and bacterial communities across broad-leaved trees and coniferous species mixed forests at the phylum level, finding that DOC was the predominant factor that significantly affected the soil bacterial communities. The soil TP, AP, and TK also showed correlations with the first two RDA axes in our study. These findings suggest that soil bacterial community composition at the OTU, species, and phylum levels in bamboo and broad-leaved mixed forests with different mixed tree species and mixing ratios was sensitive to the changes in soil properties.

**5. Conclusions**

In bamboo and single broad-leaved tree mixed forests, compared with AFB and CCB, the soil nutrient content in CAB was highest. The influence of mixed broad-leaved tree species on the diversity of soil bacterial communities was greater than that on richness, the diversity of soil bacterial communities was the highest in CAB stands with mixing ratios of 10%–20% and 20%–40%, and the dominant phyla were *Acidobacteria*, *Proteobacteria*, and *Actinobacteria*, which showed high relative abundance in CAB stands. Changes in microbial communities were associated with the availability of soil nutrients (SOC, DOC, AK, TK, $NH_4^+$, TP). In addition, the mixed broad-leaved tree species significantly changed the relationship between soil bacteria in the forest. Compared with AFB and CCB stands, the

stability of the soil bacterial community was the highest in CAB stands. The results clearly demonstrate that the mixing ratio (as represented by the width and size of the broad-leaved tree crown over the plot area) and mixed tree species affected both soil properties and community diversity and composition in bamboo broad-leaved mixed forests. The mixed tree species was the primary factor shaping the soil bacterial community composition, especially in intermediate-mixing-ratio forests, followed by soil parameters. Therefore, to promote the sustainable management of bamboo broad-leaved mixed forests, it is advisable to adjust the crown width of broad-leaved trees and select advantageous accompanying tree species to improve the soil bacterial community status of forest stands.

**Supplementary Materials:** The following supporting information can be downloaded at: https://www.mdpi.com/article/10.3390/f15060921/s1, Table S1: Characteristics of Bamboo–*Alniphyllum fortune* Hemsl. Makino Mixed Forest (AFB) sites; Table S2: Characteristics of Bamboo–*Choerospondias axillaris* Mixed Forest (CAB) sites; Table S3: Characteristics of Bamboo–*Castanopsis chinensis* Hance Mixed Forest (CCB) sites; Table S4: The relative abundant of dominant bacteria in bamboo and single broad-leaved tree mixed forest at the level of phylum

**Author Contributions:** M.Z.: Conceptualization, Writing—original draft. F.G. and S.F.: Conceptualization, Formal analysis. X.Z.: Writing—review and editing, Investigation. All authors have read and agreed to the published version of the manuscript.

**Funding:** This study was supported by the National Key R&D Program of China (2023YFD2201203), The Education Natural Science Key Project of Anhui Provice, China (2022AH051953) and the Research Start-Up Project of Huang Shan University (2022xkjq013).

**Data Availability Statement:** The data presented in this study are available on request from the corresponding author due to the data also forms part of an ongoing study.

**Acknowledgments:** We thank our colleague for reviewing our manuscript. The specific information of the reviewer is as follows: Ye Zhang (zhangye411@163.com), College of Life and Environmental Science, Huangshan University, Huangshan, Anhui, China.

**Conflicts of Interest:** The authors declare no conflicts of interest.

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
