# Peer review of "Response of Soil Microbial Community Structure and Diversity to Mixed Proportions and Mixed Tree Species in Bamboo–Broad-Leaved Mixed Forests"

_forests, doi:10.3390/f15060921_

Round 1

Reviewer 1 Report

Comments and Suggestions for Authors

The current manuscript studied the role of different broadleaved tree species and their mixing ratios in bamboo-broadleaf mixed forests, and evaluated soil microbial community structure and diversity. This study really enhances our knowledge about the role of tree species in soil microbial community diversity and therefore has novelty and potential for publication after addressing the following issues.

1.       The abstract sounds good and informative.

2.       Introduction section is well written with enough background account.

3.       Material and methods section is well drafted and provide clear description of study sites and methods used.

4.       Results section is fine with nice representation of findings.

5.       Discussion:

·         writing style and discussion of results can be improved. I came across a few lengthy and complex statements that need to modify. For example:

·         Line 332-334 “Bamboo and broadleaved mixed forest with different tree species and mixed ratio significantly aiters the soil microbial community complexity, and the influence on soil bacterial community diversity was significantly greater than the influence on its richness”.

·         Line 335-337 “The highest soil bacteria diversity were observed in CAB stand with mixed ratio 10%-20%335 20%-40%, and the lowest was soil bacterial richness were observed in CCB stand with mixed ratio 20%-40%”.

·         Section 4.2 Effects of mixed tree species and mixed ratio on the soil bacterial community

·         Line 331-352. I suggest revising this section, authors have used very general statements rather than discussing the actual cause behind current findings. Furthermore, use a simple and lucid writing style to improve audience readability.

·         Check the manuscript thoroughly for language style and grammatical errors.

·         Overall, discussion needs to be more specific and conclusive rather than justifying general statements. Also, add some latest literature. 

Comments on the Quality of English Language

Correction of Grammatical errors and English writing style need to be improved.

Author Response

Response letter

Manuscript ID forests-3013056

Paper title: Response of soil microbial community structure and diversity to mixed proportion and mixed tree species in bamboo broad-leaved mixed forest

Dear Reviews:

We would like to thank you for your efforts in reviewing our manuscript and providing many helpful comments and suggestions. Those comments are all valuable and very helpful for revising and improving our paper, as well as the important guiding significance to our researches. We have studied comments very carefully. Based on your criticisms, comments and suggestions, we have revised the manuscript accordingly. The details are explained below. Should you have any questions, please contact us without hesitate.

Response to Referees  

we would like to thank the reviewer great efforts in reading our manuscript and for your constructive comments and suggestions. Our response to the comments and suggestions are listed as follows:

Comment 1: The current manuscript studied the role of different broadleaved tree species and their mixing ratios in bamboo-broadleaf mixed forests, and evaluated soil microbial community structure and diversity. This study really enhances our knowledge about the role of tree species in soil microbial community diversity and therefore has novelty and potential for publication after addressing the following issues.

  1. The abstract sounds good and informative.
  2. Introduction section is well written with enough background account.
  3. Material and methods section is well drafted and provide clear description of study sites and methods used.
  4. Results section is fine with nice representation of findings.

ResponseWe are very grateful for your kind appraisal of Abstract, Introduction, Material and methods, Results section of the Manuscript. Thank you for your time and dedication in reviewing our manuscript. We are honored to have had the opportunity to receive feedback from someone with your level of expertise.

Comment 1: 5. Discussion:

  • writing style and discussion of results can be improved. I came across a few lengthy and complex statements that need to modify. For example:
  • Line 332-334 “Bamboo and broadleaved mixed forest with different tree species and mixed ratio significantly aiters the soil microbial community complexity, and the influence on soil bacterial community diversity was significantly greater than the influence on its richness”.
  • Line 335-337 “The highest soil bacteria diversity were observed in CAB stand with mixed ratio 10%-20%335 、20%-40%, and the lowest was soil bacterial richness were observed in CCB stand with mixed ratio 20%-40%”.

Section 4.2 Effects of mixed tree species and mixed ratio on the soil bacterial community

  • Line 331-352. I suggest revising this section, authors have used very general statements rather than discussing the actual cause behind current findings. Furthermore, use a simple and lucid writing style to improve audience readability.

         Check the manuscript thoroughly for language style and grammatical errors.

  • Overall, discussion needs to be more specific and conclusive rather than justifying general statements. Also, add some latest literature.

ResponseThanks for the referee’s kind suggestion.

First of all, Regarding the detailed discussion you mentioned, we have made detailed modifications to lines 331-352 in the manuscript you mentioned. For example:

Change “Bamboo and broadleaved mixed forest with different tree species and mixed ratio significantly aiters the soil microbial community complexity, and the influence on soil bacterial community diversity was significantly greater than the influence on its richness” to “In bamboo and broadleaf mixed forests, the diversity and richness of soil microbial community structure varied with different mixed forests under different mixed ratios. In this study, the ACE,Chao1 indexes revealed differences in the diversity and richness of soil microbial community structure across the three forest types, namely a richer bacterial community in CAB forest with 10–20%, 20%-40% mixed ratio than in other forest; the lowest value was in CCB forest with 20–40% mixed ratio.”

More details have been added to pages 4 line 360-378 of the revised manuscript.

Second, I'm sorry my English is confusing you, according to your comment, we have check all the manuscript and modified the English language, and some of the latest literature has also been added to the revised manuscript. For example:

“Ni, H., & Su, W. 2024. Spatial distribution of fine root traits in relation to soil properties and aggregate stability of intensively managed Moso bamboo (Phyllostachys edulis) plantations in subtropical China. Plant and Soil, 1-17.”

“Luo W, Zhang Q, Wang P, Luo J, She C, Guo X, Yuan J, Sun Y, Guo R, Li Z, Liu J, Tao J. 2024. Unveiling the impacts moso bamboo invasion on litter and soil properties: A meta-analysis. Sci Total Environ. 20;909:168532.”

Finally, we would like to thank the referees again for the careful reading of our paper. In addition, we have revised the manuscript carefully and believe that the new version is much better than the old one. Hope the revised version is acceptable.

Best regards

Sincerely yours

Dr. Zhang

Reviewer 2 Report

Comments and Suggestions for Authors

In the present article, microbial communities of mixed broad-leaved bamboo forests are proposed for consideration, depending on the ratio of vegetation species. The results obtained are of scientific interest for revealing the connections between the composition of vegetation, the properties of the soils under it and the microbial communities of these soils. The authors presented extensive analytical material, which, in my opinion, however, needs to be analyzed in more detail in order to better understand the relationships obtained. In addition, there are several comments on the text of the work:
Section 2.2. Specify the absolute height of the studied sites. Specify how far the sites are from each other. Give a detailed description of the soil
section 3.1. Please describe in more detail the results of the chemical analysis of the soil
line 218: It is necessary to make the table readable, possibly divided by sample types. The data is very poorly perceived.
Section 4.3. Conclusions should be rewritten in accordance with the hypotheses in lines 103-109.
Unfortunately, the paper does not consider the features of fungal communities.

Author Response

Response letter

Manuscript ID forests-3013056

Paper title: Response of soil microbial community structure and diversity to mixed proportion and mixed tree species in bamboo broad-leaved mixed forest

Dear Reviewer:

We would like to thank you for your efforts in reviewing our manuscript and providing many helpful comments and suggestions. Those comments are all valuable and very helpful for revising and improving our paper, as well as the important guiding significance to our researches. We have studied comments very carefully. Based on your criticisms, comments and suggestions, we have revised the manuscript accordingly. The details are explained below. Should you have any questions, please contact us without hesitate.

Response to Referees

we would like to thank the reviewer great efforts in reading our manuscript and for your constructive comments and suggestions. Our response to the comments and suggestions are listed as follows:

Reviewer #1:

Comment 1: In the present article, microbial communities of mixed broad-leaved bamboo forests are proposed for consideration, depending on the ratio of vegetation species. The results obtained are of scientific interest for revealing the connections between the composition of vegetation, the properties of the soils under it and the microbial communities of these soils. The authors presented extensive analytical material, which, in my opinion, however, needs to be analyzed in more detail in order to better understand the relationships obtained.

Response: Thanks for the referee’s kind suggestion. Regarding the detailed analysis you mentioned, we have modified the methods in the revised manuscript.

More details have been added to pages 4 line 142-156 of the revised manuscript. Again thanks for your carefulness and tolerance. Your comments are all valuable and very helpful for revising and improving our paper.

Comment 2: Section 2.2. Specify the absolute height of the studied sites. Specify how far the sites are from each other. Give a detailed description of the soil.

Response: Thanks for your kind advice. We are sorry for the studied sites you mentioned. First of all, in our study, the absolute height of the studied sites were between 710-816m, the slope direction of studied sites were sunny. Secondly, the minimum distance between plots was 50 m. According to your comments, we have modified it in pages 4 line 142-144 of the revised manuscript and added sample site information, as shown in Supplementary Table 1, 2, and 3). Regarding The description of the soil you mentioned, more details have been added to pages 4 line 150-156 of the revised manuscript.

Comment 3: section 3.1. Please describe in more detail the results of the chemical analysis of the soil.

Response: Thanks very much for your kind advice on detail the results of the chemical analysis of the soil. According to your comment, we have conducted a further detailed analysis of the results of physical and chemical properties in section 3.1. Which can be found on pages 6 line 213-224 of the revised manuscript.

Comment 4: line 218: It is necessary to make the table readable, possibly divided by sample types. The data is very poorly perceived.

Response: Thanks very much for your kind advice on related issues that appear in this article. Your comments have greatly helped our article. According to your comment, we provide supplementary table materials for understanding the charts. all data of the dominant bacteria were presented in Supplementary Table 4.

Supplementary Table 4 The community composition of dominant bacteria in bamboo and single broad-leaved tree mixed forest at the level of phylum

phylum

CCB

CAB

AFB

0-10%

10%-20%

20%-40%

0-10%

10%-20%

20%-40%

0-10%

10%-20%

20%-40%

Acidobacteria

0.38708

0.35153

0.37169

0.36134

0.33748

0.30519

0.34475

0.29564

0.35823

Proteobacteria

0.27204

0.27044

0.2748

0.30958

0.28612

0.27993

0.25024

0.25685

0.27656

Actinobacteria

0.12403

0.11276

0.13552

0.1026

0.13093

0.15979

0.11939

0.12401

0.13183

Chloroflexi

0.07008

0.06441

0.04526

0.05442

0.09274

0.08678

0.09992

0.09097

0.07639

Planctomycetes

0.06948

0.09094

0.08218

0.07546

0.06763

0.04658

0.08221

0.06385

0.04763

Verrucomicrobia

0.04606

0.07396

0.05856

0.06801

0.03779

0.04755

0.06219

0.0959

0.05422

WPS-2

0.01003

0.01394

0.0163

0.01139

0.01266

0.02315

0.01101

0.01035

0.01435

Gemmatimonadetes

0.00611

0.00606

0.00365

0.00492

0.01207

0.01708

0.01055

0.0153

0.01243

Firmicutes

0.00423

0.00296

0.00412

0.00249

0.00414

0.0088

0.00405

0.02045

0.00709

Bacteroidetes

0.00397

0.0022

0.00227

0.00203

0.00485

0.00508

0.00359

0.00326

0.00403

Others

0.0069

0.01081

0.00565

0.00776

0.01358

0.02006

0.01211

0.02342

0.01724

Comment 5:Section 4.3. Conclusions should be rewritten in accordance with the hypotheses in lines 103-109.

Response: Thanks for the referee’s kind advice. Based on your comments, I have made modifications to the conclusion section, which can be found on pages 16-17 line 447-463 in the revised manuscript. Special thanks to you for your good comments.

Comment 6:Unfortunately, the paper does not consider the features of fungal communities.

Response: Thanks for the referee’s kind advice. In this study, we mainly focused on soil bacteria. Just like what the referee said that the features of fungal communities not consider. Due to your suggestions, we will conduct research on fungal communities in our future work, and get more results.

Finally, we would like to thank the referees again for the careful reading of our paper. In addition, we have revised the manuscript carefully and believe that the new version is much better than the old one. Hope the revised version is acceptable.

Best regards

Sincerely yours

Dr. Zhang

Reviewer 3 Report

Comments and Suggestions for Authors

The manuscript submitted for review (Forests-3013056) is of particular scientific and practical interest.

However, I have several questions and comments.

1.       Abstract design is not the accordance with the journal requirements. The information in this section needs to be corrected.

2.       Line 95-109. The information does not correspond to the «Introduction».

3.       In «Introduction», authors should formulate the aim of the research. It is necessary to show the originality and actuality of the research.

4.       How many years did the authors do the research? Is there a difference in results by year of the study?

5.       How do the types of the tree influence the agrochemical composition of the soil? Please provide this information in the «Results» and «Discussion». Please show the correlation between soil agrochemical composition and soil microbiota.

6.       The conclusion should be re-presented, taking into account the purpose of the study, the results of the research and the perspective of using the experimental data in practical and theoretical aspects

Author Response

Response letter

Manuscript ID forests-3013056

Paper title: Response of soil microbial community structure and diversity to mixed proportion and mixed tree species in bamboo broad-leaved mixed forest

Dear Reviewer:

We would like to thank you for your efforts in reviewing our manuscript and providing many helpful comments and suggestions. Those comments are all valuable and very helpful for revising and improving our paper, as well as the important guiding significance to our researches. We have studied comments very carefully. Based on your criticisms, comments and suggestions, we have revised the manuscript accordingly. The details are explained below. Should you have any questions, please contact us without hesitate.

Response to Referees

we would like to thank the reviewer great efforts in reading our manuscript and for your constructive comments and suggestions. Our response to the comments and suggestions are listed as follows:

Reviewer #2: The manuscript submitted for review (Forests-3013056) is of particular scientific and practical interest.

However, I have several questions and comments.

Comment 1: Abstract design is not the accordance with the journal requirements. The information in this section needs to be corrected..

Response 1: Thanks for the referee’s kind advice. Based on your comments, I have made modifications to Abstract section the accordance with the journal requirements.

Comment 2: Line 95-109. The information does not correspond to the «Introduction».

Response 2: Thanks for the referee’s kind advice. Based on your comments, I have made modifications to the introduction section you mentioned, which can be found on pages 3 line 95-108 in the revised manuscript. Special thanks to you for your good comments.

Comment 3: In «Introduction», authors should formulate the aim of the research. It is necessary to show the originality and actuality of the research.

Response 3: Thanks very much for your kind advice on Introductionin this article. I have made modifications to the introduction section you mentioned. For example, “The aim of this study was to investigate the effect of mixed broad-leaved tree species and mixed ratio on soil properties, soil microbial communities, soil bacterial diversity in bamboo and broad-leaved mixed forests.We hypothesized that (1) AFB will have a greater adverse effect on soil nutirent and microbial structure than CAB, and intermediate in CCB forests. Further, we hypothesized that (2) microbial structure and diversity will be strongly impacted by mixed forest types at intermediate mixed ratio forests, but less at lower mixed ratio forests, we also hypothesized that (3) the mixed proportion and mixed species of bamboo and broad-leaved mixed forest affect the microbial community structure and diversity by influencing soil nutrient status. We assessed the impact of tree species and mixing ratios on soil microbial communities by measuring soil properties and the soil microbial communities diversity and composition.Three bamboo and broad-leaved tree mixed forests were  designed to put those hypotheses to the test including bamboo-Castanopsis chinensis Hance mixed forest (CCB), bamboo-Alniphyllum fortune (Hemsl.) Makino mixed forest (AFB), and bamboo-Choerospondias axillaris mixed forest(CAB) was designed to put those hypotheses to the test.” More solutions have been added into the revised manuscript which can be found on pages 2 and 3 in the revised manuscript.

Comment 4: How many years did the authors do the research? Is there a difference in results by year of the study?

Response 4: Unfortunately, we only conducted one year of research.  We will continue to monitor it for many years in the future.

Comment 5: How do the types of the tree influence the agrochemical composition of the soil? Please provide this information in the «Results» and «Discussion». Please show the correlation between soil agrochemical composition and soil microbiota.

Response 5: Thanks very much for your kind advice on related issues that appear in this article. Your comments have greatly helped our article.  Frist of all, according to your comment, we have provided detailed description of the impact of tree types on the agricultural chemical composition of soil the «Results» and «Discussion», more details have been added to pages 6 line 213-224 and pages14 line 330-358 of the revised manuscript.

Second, the correlation between soil agrochemical composition and soil microbiota were described in pages 16 line 434-445 of the revised manuscript.

Special thanks to you for your good comments.

Comment 6:The conclusion should be re-presented, taking into account the purpose of the study, the results of the research and the perspective of using the experimental data in practical and theoretical aspects.

Response 6: Thanks for the referee’s kind advice. Based on your comments, I have made modifications to the conclusion section, which can be found on pages 16-17 line 447-463 in the revised manuscript. Special thanks to you for your good comments.

Finally, we would like to thank the referees again for the careful reading of our paper. In addition, we have revised the manuscript carefully and believe that the new version is much better than the old one. Hope the revised version is acceptable.

Best regards

Sincerely yours

Dr. Zhang

Round 2

Reviewer 1 Report

Comments and Suggestions for Authors

I am satisfied with the author's response. 

Reviewer 3 Report

Comments and Suggestions for Authors

The authors have carefully attended the valuable feedback provided by the reviewers , therefore, the publication may be accepted.

However, the finishing decision should be given by the editor of the journal and the editor of the special issue on the publication of this manuscript since the experimental data were obtained for only one growing season.  Does this journal accept manuscripts with one-year data?